# Genome-Wide Identification and Expression Pattern of the *GRAS* Gene Family in Pitaya (*Selenicereus undatus* L.)

**DOI:** 10.3390/biology12010011

**Published:** 2022-12-21

**Authors:** Qamar U Zaman, Muhammad Azhar Hussain, Latif Ullah Khan, Jian-Peng Cui, Liu Hui, Darya Khan, Wei Lv, Hua-Feng Wang

**Affiliations:** 1Hainan Yazhou Bay Seed Laboratory, Sanya Nanfan Research Institute, Hainan University, Sanya 572025, China; 2College of Tropical Crops, Hainan University, Haikou 570228, China

**Keywords:** genome-wide analysis, *GRAS* gene family, growth and development, pitaya/dragon fruit, *S. undatus* L.

## Abstract

**Simple Summary:**

The *GRAS* gene family plays a critical role in regulation of growth, defense, light and hormone responses. We identified 45 *GRAS* genes in the pitaya genome and categorized them into nine respective subfamilies: PAT1, SHR, LISCL, HAM, SCR, RGL, LAS, DELLA and SCL-3. Among these 45 *HuGRAS* family members, we reported nine candidate genes that played key roles in the growth and development of the pitaya plant.

**Abstract:**

The *GRAS* gene family is one of the most important families of transcriptional factors that have diverse functions in plant growth and developmental processes including axillary meristem patterning, signal-transduction, cell maintenance, phytohormone and light signaling. Despite their importance, the function of *GRAS* genes in pitaya fruit (*Selenicereus undatus* L.) remains unknown. Here, 45 members of the *HuGRAS* gene family were identified in the pitaya genome, which was distributed on 11 chromosomes. All 45 members of *HuGRAS* were grouped into nine subfamilies using phylogenetic analysis with six other species: maize, rice, soybeans, tomatoes, *Medicago truncatula* and Arabidopsis. Among the 45 genes, 12 genes were selected from RNA-Seq data due to their higher expression in different plant tissues of pitaya. In order to verify the RNA-Seq data, these 12 *HuGRAS* genes were subjected for qRT-PCR validation. Nine *HuGRAS* genes exhibited higher relative expression in different tissues of the plant. These nine genes which were categorized into six subfamilies inlcuding DELLA (*HuGRAS-1*), SCL-3 (*HuGRAS-7*), PAT1 (*HuGRAS-34*, *HuGRAS-35*, *HuGRAS-41*), HAM (*HuGRAS-37*), SCR (*HuGRAS-12*) and LISCL (*HuGRAS-18*, *HuGRAS-25*) might regulate growth and development in the pitaya plant. The results of the present study provide valuable information to improve tropical pitaya through a molecular and conventional breeding program.

## 1. Introduction

Pitaya fruit, also known as dragon fruit, belongs to the Cactaceae family, which comprises 127 genera and 1750 species [1]. Among these species, pitaya-*Selenicereus undatus* (*S. undatus*) formerly known as *Hylocereus undatus* (2n = 2x = 22) is a diploid perennial climbing plant that originated from rainforests in the tropical and subtropical regions of Mexico and Colombia [2]. Pitaya fruit gained the attention of growers due to its attractive fuchsia color, its delicious aroma and its ability to tolerate harsh environmental conditions [1,3]. It contains vitamin C and has high antioxidant properties linked to its phenolic and betacyanin content [4]. Several transcription factors (TFs), including *WRKY* [5], *MYB* [6], *MADS*-box [7], *ARF* [8], *AP2*/*EREBP* [9], *HB* [10], *SBP* [11], *bZIP* [12], APX [13] and the *GRAS* family, are being explored to identify their specific roles in plants [14]. Significant research has been conducted on the *GRAS* gene family in many crops, including *Arabidopsis thaliana* [15], Chinese cabbage [16], switchgrass [17] *Medicago sativa* [18], cassava [19], maize [20,21], rice [22], *Melilotus albus* [23], wheat [24], canola [25], foxtail millet [26] and soybean [27], but rarely in tropical fruits such as litchi [28]. To improve the growth and developmental process, it is important to report the *GRAS* gene family in pitaya fruit.

TFs are proteins that contain domains that bind to the promoter regions that transcribe DNA into mRNA. The *GRAS* TF is named after the first reported TFs: gibberellic acid insensitive (GAI) [29,30], REPRESSOR of GAI (RGA) [31] and SCARECROW (SCR) [32]. GRAS proteins consist of 360–850 amino acids, C-terminal homology and five carboxyl-terminal motifs with the same sequence in the whole family [30,33,34]. The *GRAS* protein can be divided into five peptide regions, and it carries highly conserved motifs in a specific order: leucine heptad repeat I (LHRI), VHID, leucine heptad repeat II (LHRII), PFYRE and SAW [34,35]. The *GRAS* gene family is involved in multiple phytohormone-signaling pathways and plays a diverse role in signal transduction [35,36], root patterning [37], meristem formation, shoot development [35], axillary meristem patterning and cell maintenance.

The *GRAS* gene family is divided, based on its structure, into the following subfamilies: DELLA, HAIRY MERISTEM (HAM), LATERAL SUPRESSOR (LAS), *Lilium longiflorum* SCARECROW-LIKE (LISCL), REPRESSOR OF GAI-LIKE (RGL), PHYTOCHROME A SIGNAL TRANSDUCTION 1 (PAT1), SCARECROW (SCR), SCARECROW-LIKE 3 (SCL3) and SHORT ROOT (SHR) [35,38,39,40]. *GRAS* subfamilies perform transcriptional regulation and are involved in specific functions, as the DELLA subfamily negatively interacts with the gibberellic-acid (GA) and light-signaling pathways [41], HAM with shoot-stem-cell initiation and proliferation [42] and LAS with axillary meristems [43,44]. The LISCL/TGA complex responds to defense and stress tolerance, PAT1 interacts with phytochrome-A signal transduction [45] and SCR interacts with radial patterning in roots and shoots [46]. SCL3 acts as an attenuator of DELLA proteins and represses their expression antagonistically [47], and SHR interacts with endodermis specification and root patterning [37]. Through genome-wide analysis, different *GRAS* genes have been predicted in different crop species: 48 in Chinese cabbage [16], 144 in switchgrass [17], 87 in canola (*Brassica napus*) [25], 59 in *Medicago truncatula* [48], 52 in quinoa [49], 117 in soybeans (*Glycine max* L.) [21], 150 in cotton (*Gossypium hirsutum*) [50], 62 in barley (*Hordeum vulgare*) [51], 37 in bottle gourds [52], 50 in sweet oranges [53], 48 in litchi (*Litchi chinesis* Sonn) [28], 57 in rice (*Oryza sativa* L.) and maize (*Zea mays* L.) [30], 55 in *Melilotus albus* [23], 50 in pepper (*Capsicum annuum* L.) [54] and 32 in *Arabidopsis thaliana* [55].

In this study, we performed a comprehensive genome-wide analysis of the *GRAS* gene family and identified 45 *GRAS* members in the pitaya (*S. undatus L*.) genome, mapped to 11 chromosomes (chrs). The *GRAS* genes were identified, and phylogenetic relationships were established with the previously reported *GRAS* proteins of maize (*Zea mays* L.), soybeans (*Glycine max* L.), *Mediacgo trunctula*, rice (*Oryza sativa* L.), *Arabidopsis thaliana* and tomatoes (*Solanum lycopersicum*). We identified the locations of 45 genes on the chromosome and selected 12 genes to check their expression patterns in different tissues of the pitaya plant. Among them, *HuGRAS-1*, *HuGRAS-6*, *HuGRAS-12*, *HuGRAS-18*, *HuGRAS-25*, *HuGRAS-34*, *HuGRAS-35*, *HuGRAS-37* and *HuGRAS-41* were found to be important genes that exert their potential functions in the growth and development of the pitaya (*S. undatus* L.) plant.

## 2. Materials and Methods

### 2.1. Retrieval of GRAS Family Members in Pitaya

The genome of the pitaya plant (*S. undatus* L.) was downloaded from the pitaya genome database, http://www.pitayagenomic.com/ (accessed on 1 September 2022) [56]. All previously published information about *GRAS* proteins was retrieved from the NCBI website, https://www.ncbi.nlm.nih.gov/protein (accessed on 24 September 2022), and Phytozome, https://phytozome-next.jgi.doe.gov/ (accessed on 2 October 2022) [57]. The InterPro tool, https://www.ebi.ac.uk/interpro/ (accessed on 29 November 2022), was used to find the domains of the *GRAS* proteins. The characterized protein sequences of corn (*Zea mays* L.) [30], soybeans (*Glycine max* L.) [21], *Mediacgo truncatula* [48], rice (*Oryza sativa* L.) [55], *Arabidopsis thaliana* [46] and tomatoes (*Solanum lycopersicum*) [58] were obtained from previous studies. *GRAS*-protein physical and chemical properties, including each protein’s molecular weight, isoelectric point and grand average of hydropathicity (GRAVY), were computed using the Expasy ProtParam Tool, https://web.expasy.org/protparam/ (accessed on 9 October 2022).

### 2.2. Domain Analysis of HuGRAS Proteins

All 45 *HuGRAS* protein sequences were subjected to finding of conserved domains using the NCBI conserved domain tool, https://www.ncbi.nlm.nih.gov/Structure/cdd/wrpsb.cgi (accessed on 13 October 2022), against Pfamv34.0-19178pSSMs. The retrieved data was used to draw the structure of the *GRAS* domain using TBTools [59]. A motif finder, https://www.genome.jp/tools/motif/ (accessed on 17 October 2022), was also used to compute the concerned motifs.

### 2.3. Phylogenetic Analysis of HuGRAS Family

All *GRAS* protein sequences of Arabidopsis (*A. thaliana* araport11), *Medicago truncatula* (*Medicago truncatula* Mt4.0v1—barrel medic), soybeans (*Glycine max* Wm82.a4.v1), rice (*Oryza sativa*—v7.0), tomatoes (*Solanum lycopersicum*—ITAG4.0) and corn (*Zea mays*—Refgen_V4) were downloaded from Phytozome-13 (https://phytozome-next.jgi.doe.gov (accessed on 13 October 2022)) [57], and pitaya (*S. undatus*—Guanhuabai) protein sequences were retrieved from the “pitaya-genome-website” [56]. All 380 protein sequences were used to perform alignment through molecular evolutionary genetic analysis (MEGA-11) [60]. The aligned protein sequences were employed for phylogenetic analysis with the maximum likelihood tree test, which used 1000 bootstrap replicates. Finally, the tree was visualized using iTOL software [61].

### 2.4. HuGRAS Genes Distribution on Pitaya Chromosomes

All 45 *GRAS* genes’ data and genomic DNA were obtained from the pitaya genome database, http://www.pitayagenomic.com/ (accessed on 1 September 2022) [56]. Chromosome length was calculated using TBTools “FASTA stats” [59]. Then, a chr ideogram was created using the PhenoGram plot tool, http://visualization.ritchielab.org/phenograms/plot (accessed on 1 September 2022). The output image thereof was used to mention all *HuGRAS* genes on the chromosome.

### 2.5. Pattern and Distribution of Conserved Motifs

The MEME suite program was used to analyze the 45 *HuGRAS* genes and identify conserved motifs. The default parameters were used to identify the maximum 10 motifs at https://meme-suite.org/meme/tools/meme (accessed on 13 October 2022).

### 2.6. Expression Analysis of HuGRAS Genes

The expression pattern of the *HuGRAS* gene family was found via the pitaya genome database (http://www.pitayagenomic.com/ (accessed on 17 October 2022)) [56]. All 45 *HuGRAS* genes’ expression levels were determined using different tissues of the plants, including four stages of flower buds (FB1, FB2, FB3, FB4), five stages of flower (F1, F2, F3, F4, F5), three stages of pericarp (PeriC-45d, PeriC-65d, PeriC-85d) and three stages of fruit pulp (Pulp-29d, Pulp-35d, Pulp-49d).

### 2.7. Network Analysis of HuGRAS Proteins

All 45 *HuGRAS* genes were subjected to collection of their interactions with other genes using the pitaya genome website, http://www.pitayagenomic.com/coexpression (accessed on 30 October 2022) [56]. The Cytoscape tool was used to build the network using information from all of the identified and interacting genes [62].

### 2.8. Cis-Acting Element Analysis in HuGRAS Promoter Sequences

Promoter sequences were retrieved from the pitaya genome file using TBTools (2000 bp upstream of the start codon) [59]. The PlantCARE database was used to retrieve cisacting regulatory elements [63].

### 2.9. Plant Materials

The “Shuangse Dahong” pitaya variety was used in this experiment, and flower buds were collected from the germplasm resource of Hainan-Shengda Modern Agriculture Development Company, Qionghai, Hainan, China. All plants were grown in field conditions.

### 2.10. RNA Isolation and Real-Time Quantitative PCR Expression Analysis

Utilizing the RNAprep Pure Plant Kit, total RNA was extracted (TIANGEN, Beijing, China). The plant material used for RNA extraction included stems (one-month-old stems, one-year-old stems and two-year-old stems, designated S1Ms, S1Ys and S2Ys, respectively), flower buds (FBs), pericarp (PeriC) and pulp. A NanoDrop 2000C spectrophotometer was used to measure the concentration of the samples (Thermo Fisher Scientific, Waltham, MA, USA). DNase I was used to remove genomic DNA from a total of 1 g of RNA from each sample before being utilized as a template for reverse transcription to create the desired amount of cDNA (QuantiTect Reverse Transcription Kit; Qiagen, Shanghai, China). The RNA sample for each qRT-PCR was standardized using the actin-gene-expression level in *S. undatus* L. Three biological and three technical replications of each sample were used in the qPCR, with ACTIN serving as the internal control. The SYBER Green Master Mix (Novogene, Shanghai, China) was used, along with the LightCycler 480 real-time PCR system (Applied Biosystem, St. Louis, MO, USA). qRT-PCR results were analyzed using the double-delta CT method [64,65].

## 3. Results

### 3.1. Genome-Wide Identification of the GRAS Family in Pitaya

Through genome-wide analysis, 45 candidate genes were retrieved from the pitaya genome, and these genes were designated *HuGRAS*-1 to *HuGRAS*-45. Basic physical and chemical properties of the *HuGRAS* genes, including each gene’s chromosome number, position on the chromosome, CDS length, protein length, protein molecular weight, isoelectric point (pI) and GRAVY, are summarized in Table 1. Protein length and molecular weight varied greatly, ranging from 97 (*HU08G00229.1*) to 809 AA (*HU01G00472.1*), and molecular weight ranged from 15–95 kDa. All 45 pitaya *HuGRAS* proteins were predicted to be hydrophilic because their representative GRAVY values were less than 0, ranging from −0.006 (*HU06G00376.1*) to −0.436 (*HU02G01570.1*). All *HuGRAS* proteins comprised varying degrees of pI values, ranging from 5.4 (*HU10G00709.1*) to 9.5 (*HU08G00229.1*), with an average value of 7.2.

Furthermore, domain-based analysis was carried out for all 45 *HuGRAS* proteins using an NCBI domain search, and the retrieved data and TBTools were further used to draw the structure of the domain. This domain-based analysis confirmed the presence of the *GRAS* family on 45 selected protein sequences (Figure 1).

### 3.2. Phylogenetic Analysis of the GRAS Gene Family

To construct a phylogenetic tree, protein sequences from the characterized species were retrieved from previous studies, including those on Arabidopsis, *Medicago truncatula*, tomatoes, rice, soybeans and maize. Characterized *GRAS* protein sequences from six species and the Phytozome website were used to retrieve all *GRAS* proteins from their respective genomes. Collectively, 380 protein sequences were used to draw a phylogenetic tree, including the 45 *HuGRAS* protein sequences from the pitaya genome and 335 protein sequences from the other six species (Appendix A). In the resulting phylogenetic tree, the *GRAS* genes were divided into nine subfamilies: PAT1, SHR, LISCL, HAM, SCR, RGL, LAS, DELLA and SCL3 (Figure 2). All 45 *HuGRAS* proteins were grouped as follows: twelve in PAT1; ten in LISCL; five in HAM; four each in SHR, SCL3 and SCR; three in DELLA; two in LAS; and one in RGL. The PAT1 subfamily contained 12 types of *HuGRAS* protein and was the largest subfamily of *GRAS* protein, while RGL had only one type of *HuGRAS* protein and was one of the smallest subfamilies of *GRAS* protein.

### 3.3. HuGRAS-Protein Sequence Alignments and Conserved Motifs

A conserved-motif analysis of each *GRAS* protein was carried out using the MEME tool. Ten conserved motifs were identified. The C terminal regions of the *HuGRAS* proteins contained highly conserved domains. Motif 2, motif 6 and motif 7 were found in almost all of the *HUGRAS* proteins. However, motif 5 was not found in *HuGRAS-35*, *HuGRAS-39*, *HuGRAS-42* or *HuGRAS-45* (Figure 3). Most of the *GRAS* proteins carried similar motifs within the group, with very few motif differences. These findings also helped us to understand the close evolutionary relationships of the same protein group. The known-motif of the amino-acid-sequence is exhibited in Appendix A.

### 3.4. Gene Structure and Distribution of HuGRAS Genes on Chromosomes

To find the gene structures, the intron and exon structures of all of the *HuGRAS* genes were aligned (Appendix A). Among all 45 *HuGRAS* genes, most of the *HuGRAS* gene sequences showed two sequences of introns and one sequence of exons. The majority of the *HuGRAS* genes had a similar pattern of exons, indicating the phylogeny and evolution of their gene family, except for *HuGRAS-8* and *HuGRAS-42*.

The *HuGRAS* genes were physically located on 11 chrs in the pitaya genome. All *GRAS* genes were mapped on pitaya chrs based on the information available at the pitaya genome website, http://www.pitayagenomic.com/ (accessed on 1 September 2022). The chr length and position of each gene of the pitaya genome is presented in Appendix A. Forty-five *HuGRAS* genes were unevenly distributed on 11 chrs. Most of the *HuGRAS* genes were found on chr 02 and chr 08. Chr 10 had one *GRAS* gene but chr 09 had no *HuGRAS* genes (Figure 4).

### 3.5. Expression Analysis of HuGRAS Genes in Different Tissues of Pitaya

*GRAS* TFs and *GRAS* subfamily members, including DELLA, HAM, LAS, LISCL, PAT1, SCR, SCL3, SHR and RGL, play important roles in plant growth and development, axillary meristem formation, root radial patterning, cell maintenance and proliferation, defense response and stress tolerance. The genes in each tissue play central roles in pitaya development. The expression of *GRAS* genes in the pitaya plant comprises 15 tissues, including flower buds (four stages—FB1 to FB4), flowers (five stages—F1 to F5), pericarp (three stages—45 days, 65 days, 85 days) and the pulp of the fruit (three stages—29 days, 35 days, 49 days). Of the 45 *HuGRAS* genes, most were not expressed (Figure 5). We choose 12 genes that showed significant differential expression in all tissues: *HuGRAS-1*, *HuGRAS-6*, *HuGRAS-7*, *HuGRAS-12*, *HuGRAS-18*, *HuGRAS-21*, *HuGRAS-25*, *HuGRAS-29*, *HuGRAS-34*, *HuGRAS-35*, *HuGRAS-37* and *HuGRAS-41*.

### 3.6. HuGRAS Proteins Network Analysis

The *HuGRAS* proteins and their interaction network revealed that the number of proteins regulated by each predicted gene is significantly different. Among the 45 *HuGRAS* genes, 27 genes were involved in 215 possible interactions. Based on network analysis, we divided the interacting genes into four categories: gray (2–5 interactions), yellow (6–10 interactions), red (11–15 interactions) and green (16–20 interactions). Based on the maximum interaction, we identified *HuGRAS-1*, *HuGRAS-18*, *HuGRAS-6*, *HuGRAS-36* and *HuGRAS-39* as hub genes, shown with green and red color (Figure 6). In the yellow category, we found that *HuGRAS-12*, *HuGRAS-29*, *HuGRAS-35* and *HuGRAS-37* interacted significantly with other genes.

### 3.7. Identification of Cisacting Elements in HuGRAS Promoter Sequences

To identify the biological functions (stress response, growth and development) of the *HuGRAS* genes, all 45 *HuGRAS* gene sequences (2000 bp upstream of start codon) were selected for cis-element analysis using the PlantCARE web tool (Appendix A). In total, 17 cis-elements were recorded in this study (Appendix A). The cis-regulatory elements of 45 *GRAS* proteins are shown in Appendix A. Nine genes that exhibited higher expression among the *GRAS* gene family in the pitaya plant exhibited various cis-acting regulatory elements, as shown in Figure 7. Fourteen cis-acting elements were categorized into four groups: light-responsive elements, growth and development elements, stress- and defense-responsive elements and hormone-responsive elements. 

### 3.8. Expression of HuGRAS Genes at Developmental Stages of Pitaya

To confirm the expressions of the predicted genes from the transcriptome data, we conducted qRT-PCR for *HuGRAS-1*, *HuGRAS-6*, *HuGRAS-7*, *HuGRAS-12*, *HuGRAS-18*, *HuGRAS-21*, *HuGRAS-25*, *HuGRAS-29*, *HuGRAS-34*, *HuGRAS-35*, *HuGRAS-41* and *HuGRAS-37*. We designed primers for the 12 candidate genes, categorized in six *GRAS* subfamilies (Appendix A). The results thereof exhibited that *HuGRAS-1*, *HuGRAS-7*, *HuGRAS-12*, *HuGRAS-18*, *HuGRAS-25*, *HuGRAS-34*, *HuGRAS-35*, *HuGRAS-41* and *HuGRAS-37* showed higher levels of expression across the tissues (Figure 8). The expression levels of the *HuGRAS* members varied widely in different tissues. The *HuGRAS-1* gene, categorized in the DELLA subfamily, was significantly expressed across the tissues, including the stems, FBs and pericarp. However, relatively weaker expression was observed in the pulp of the fruit. Among the PAT1 subfamily members, *HuGRAS-34*, *HuGRAS-35* and *HuGARS-41* exhibited strong expression in the plant tissues as compared to *HuGRAS-6*, which exhibited weak expression in the pericarp and the pulp. *HuGRAS-7,* which belongs to the SCL-3 subfamily, was expressed at a low level in the one-month-old stem cells but abundant in other tissues. *HuGRAS-12*, a gene categorized in the SCR subfamily, was expressed at a higher level in other tissues than the flower buds. *HuGRAS-21* and *HuGRAS-29*, members of the LISCL subfamily, were expressed at lower levels than the *HuGRAS-18* and *HuGRAS-25* grouped in the same subfamily, which were expressed at higher levels in the flower buds, the pericarp and the pulp of the pitaya plant. *HuGRAS-37*, grouped into the HAM subfamily, was highly expressed in the flower buds but weakly expressed in the one-month-old stems. Nine genes, which were categorized into six subfamilies, exhibited higher expression levels and might play key roles in the growth and development of the pitaya plant.

## 4. Discussion

Pitaya (*S. undatus* L.) is a tropical fruit, typically cacti, evergreen, and consists of cladodes (a modified stem replaces the leaves for photosynthesis function) which perform its functioning as a leaf. The flowers and fruits are edible, and the pericarp and pulp of *S. undatus* are white in color. The fruits are highly enriched with polyphenols, tannis, betalains and nonbetalainic and antioxidant compounds [66]. Due to the importance of the pitaya tropical fruit, the present study was carried out to explore the growth and the developmental process of the plant. The *GRAS* TF is being explored in other crop species, such as Arabidopsis [55], *Medicago truncatula* [48], pepper [54], cotton [50], soybeans [27], tomatoes [67], Chinese cabbage [16] and tropical fruit such as litchi [28], but we could not find any research studies about the *GRAS* gene family in *S. undatus* L. *GRAS* proteins have been recognized as important TF, playing different functions in plant growth and development, including patterning of roots and shoots, responses to various kinds of stresses, stem-cell initiation and maintenance [35], light signaling and the gibberellic-acid signal-transduction pathway [21,68].

With the availability of the pitaya reference genome [2] and pitaya tissue expression data via the pitaya genome and multiomics database [56], we performed a genome-wide identification of the *GRAS* gene family members in the pitaya genome. In the current study, we found 45 *GRAS* gene family members in this genome, named *HuGRAS-1* to *HuGRAS-45*; they were widely distributed on 11 chromosomes (Figure 4). Most of the *HuGRAS* genes were found were on the ends of these chromosomes, which is in accordance with other plant species, such as watermelon, potatoes, rice and Arabidopsis [69]. The conserved motif structures (Figure 3) and *HuGRAS* gene sequences (Appendix A) exhibited the same pattern of conserved motifs and exon–intron sequences, respectively, suggesting that these genes may have similar functions to those reported in previous studies [21].

In accordance with phylogenetic analysis, we compared the 45 *HuGRAS* gene sequences with 335 sequences of *GRAS* proteins from maize, soybeans, *Medicago truncatula*, rice, Arabidopsis and tomatoes. *HuGRAS* genes were divided into nine subfamilies based on clade support values: PAT1, SHR, LISCL, HAM, SCR, RGL, LAS, DELLA and SCL3 (Figure 2). Each subfamily carried varying numbers of *HuGRAS* genes, and the PAT1 subfamily contained the largest number of *HuGRAS* genes. The protein sequences and differential expression profiles of pitaya tissues aid in the identification of the genes that play key roles in growth and development. Expression and network analysis provide a clue to locating genes that exhibit high levels of expression (Figure 5) and interact with many other genes (Figure 6). With the help of expression analysis and network analysis, 12 selected genes were categorized into their respective subfamilies, as predicted in the phylogenetic tree (Figure 2). All genes were placed in their respective *GRAS* families: *HuGRAS-1* in the DELLA subfamily; *HuGRAS-6*, *HuGRAS-34*, *HuGRAS-35* and *HuGRAS-41* in the PAT1 subfamily; *HuGRAS-7* in the SCL-3 subfamily; *HuGRAS-12* in the SCR subfamily; *HuGRAS-18*, *HuGRAS-21*, *HuGRAS-25* and *HuGRAS-29* in the LISCL subfamily; and *HuGRAS-37* in the HAM subfamily. qRT-PCR was carried out for the predicted gene subfamilies (Figure 8) to confirm their expressions in different stages of the plant. The *HuGRAS-1* gene, categorized in the DELLA subfamily, was significantly expressed across the tissues, as the DELLA subfamily is involved in the growth and development of the plant [41]. In the absence of gibberellic acid, DELLA proteins interact with light-responsive TFs, including phytochrome-interacting factors (PIFs), to form inactive complexes [70], while higher expression of gibberellic acid degrades DELLA proteins and initiates the growth rate [71]. Our network analysis revealed that the *HuGRAS-1* gene interacts with almost 20 other proteins, so our results are consistent with the prediction of Hirsch and Oldroyd, 2009 [41] that DELLA proteins interact with other PIF families and make complexes with them. This DELLA–PIF TF complex is possibly competitive but a common mechanism for DELLAs to make complexes for light- and gibberellic-acid-signaling to alter environmental conditions [41]. DELLA proteins also regulate immune responses by regulating the jasmonic- and salicylic-acid pathways. The PAT1 subfamily members, *HuGRAS-34*, *HuGRAS-35* and *HuGARS-41*, exhibited strong expressions in plant tissues as compared to *HuGRAS-6*, which was weakly expressed. PAT1 is a specific member of the *GRAS* family that interacts with light signaling via phytochrome A to regulate the plant developmental process, including de-etiolation and hypocotyl elongation [45]. *HuGRAS-7,* which belongs to the SCL-3 subfamily, was expressed at a low level in one-month-old stem cells but exhibited higher expression abundantly in other tissues. The SCL-3 subfamily acts antagonistically, downstream to gibberellic-acid DELLA responses and upstream to gibberellic-acid-biosynthesis pathways, during plant growth and development [47]. *HuGRAS-12* was categorized in the SCL-3 subfamily and was expressed at a higher level in other tissues than flower buds. *HuGRAS-37* is grouped into the HAM subfamily and was highly expressed in flower buds but weakly expressed in one-month-old stems. The SCR and HAM subfamilies play key roles in root/shoot patterning and cell differentiation in shoot meristem maintenance [72]. *HuGRAS-21* and *HuGRAS-29*, members of the LISCL subfamily, were expressed at lower levels than the *HuGRAS-18* and *HuGRAS-25* genes of the same subfamily, which were expressed at a higher level in the flower buds, the pericarp and the pulp of the pitaya plant. Higher levels of expression of LISCL genes (*HuGRAS-18* and *HuGRAS-25*) may predict their role in flower development and fruit ripening. The LISCL subfamily of the *GRAS* protein has been reported to play a role in another development, of *Lilium longiflorum* L. [73]. Previous studies have revealed that redundancy of relative expression and phytohormones in different parts of the reproductive tissue (panicle) can lead to defects in growth. Similarly, varying expression levels of GRAS-family genes might also have different functions in different pitaya tissues [74,75].

In this study, we analyzed the *GRAS* TF family in the pitaya plant (*S. undatus* L.) and six other species, including maize, soybeans, *Medicago truncatula*, rice, Arabidopsis and tomatoes. A total of 380 *GRAS* genes were analyzed in this research, in addition to 45 genes that were predicted from the pitaya genome. We categorized these genes into nine subfamilies based on phylogenetics and previous studies of other crops. Among the nine subfamilies of *GRAS*, few genes showed higher expression in different tissues of pitaya plant. These genes were categorized into six sub-families including DELLA (*HuGRAS-1*), SCL-3 (*HuGRAS-7*), PAT1 (*HuGRAS-34*, *HuGRAS-35*, *HuGRAS-41*), HAM (*HuGRAS-37*), SCR (*HuGRAS-12*) and LISCL (*HuGRAS-18*, *HuGRAS-25*) which may have potential key role in the growth and development of the pitaya plant. Their roles were also confirmed using in silico cis-acting analysis (Appendix A). As we could see, cis-acting elements, including gibberellin, auxin, ABA, jasmonic-acid and salicylic-acid-responsive elements were abundantly present in the *HuGRAS* promoters. These genes can be used to study the regulatory pathways of specific plant traits. Positive and negative regulators can be identified from the pathways; then the CRISPR system can be used to produce a transgene-free pitaya plant. Previously, many crops have been improved using the latest genome editing technique [76,77]. Collectively, our results lay a theoretical foundation for the role of *GRAS* genes in pitaya growth and development. It provides valuable information to improve the pitaya breeding program.

## 5. Conclusions

This study is the first comprehensive genome-wide identification of the *GRAS* gene family in pitaya (*S. undatus* L.). This research might aid in the interpretation of the *GRAS* genes function, protein interactions, signaling-pathway regulations and expression patterns in different tissues. The comparative study between the *GRAS* families of six species, the phylogenetic tree, the expression pattern and the gene network analysis will lay a foundation for the functional characterization of the genes in pitaya. Understanding the possible roles of nine predicted genes (*HuGRAS-1*, *HuGRAS-7*, *HuGRAS-12*, *HuGRAS-18*, *HuGRAS-25*, *HuGRAS-34*, *HuGRAS-35*, *HuGRAS-37*, *HuGRAS-41*) from the six subfamilies of *GRAS* gene and their expression patterns in different tissues provides insightful information for the development of pitaya fruit’s economic, agronomic and ecological benefits. Altogether, the current study is the first report on the *GRAS* gene family in pitaya tropical fruit. The identification of the genes will assist in clarifying the molecular genetic basis and aid in improving the genotypes in the breeding program.

## Figures and Tables

**Figure 1 biology-12-00011-f001:**
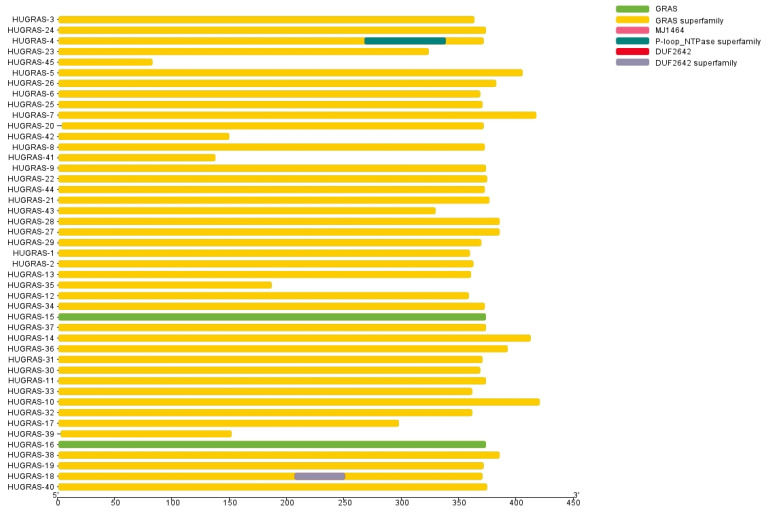
*GRAS*-family protein domains. All 45 *HuGRAS* sequences (*HuGRAS*-1 to *HuGRAS*-45) contained *GRAS* domains.

**Figure 2 biology-12-00011-f002:**
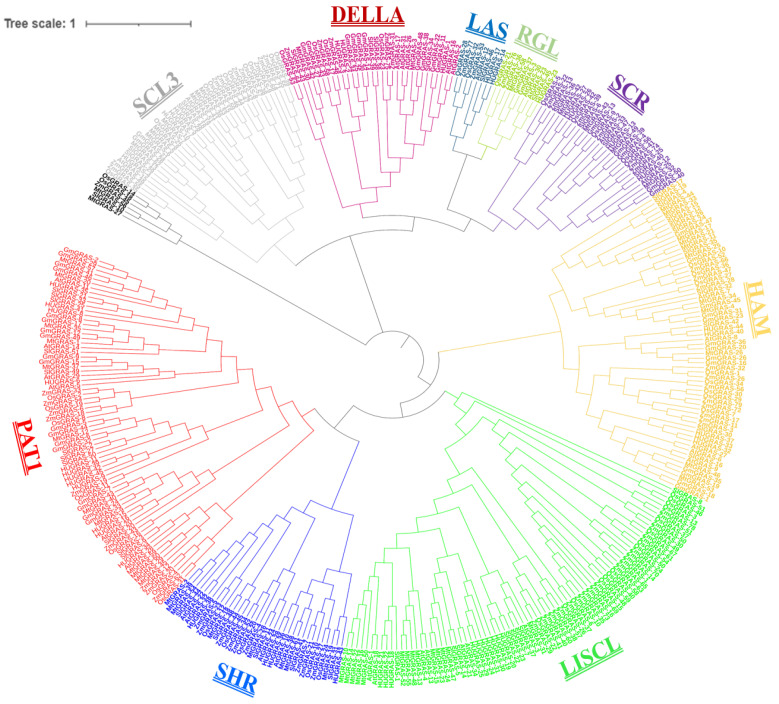
Characterized sequences of six species (maize, soybeans, Arabidopsis, *Medicago truncatula*, tomatoes and rice) were used to draw this phylogenetic tree with the pitaya *GRAS* genes. The *GRAS* proteins were divided into nine subfamilies, exhibited with different colors: PAT1 (red), SHR (navy blue), LISCL (lime), HAM (light orange/wheat color), SCR (violet), RGL (pea green), LAS (teal), DELLA (magenta pink) and SCL3 (gray).

**Figure 3 biology-12-00011-f003:**
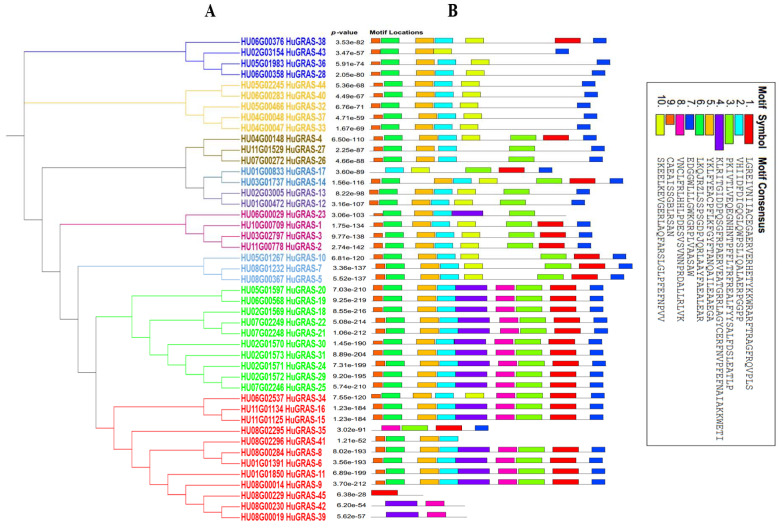
Distribution of putative motifs in each *HuGRAS* protein sequence. (**A**) The rectangular phylogenetic tree of 45 *HuGRAS* proteins was constructed using MEGA-11 software based on the maximum likelihood method, with a bootstrap value of 1000 replicates. (**B**) Conserved motifs of pitaya, named *HuGRAS*-1 to *HuGRAS*-45, that were predicted using the MEME program and plotted in TBTools software. Motif 1 to motif 10 are shown in differently colored boxes.

**Figure 4 biology-12-00011-f004:**
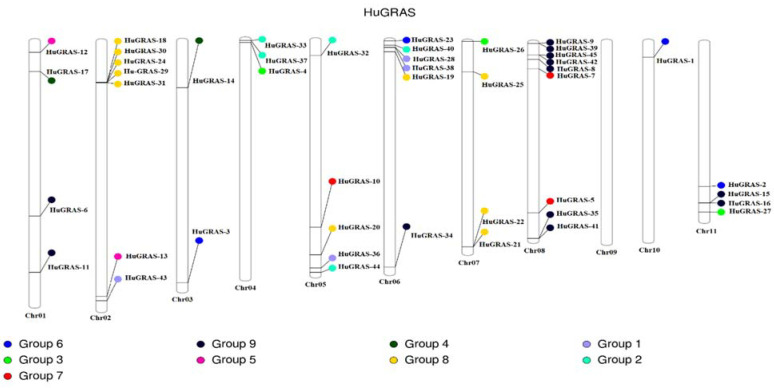
Distribution of 45 *HuGRAS* genes on 11 pitaya (*S. undatus* L.) chromosomes. Gene names are mentioned in black. *HuGRAS* genes are divided into nine groups on the basis of their domain structures.

**Figure 5 biology-12-00011-f005:**
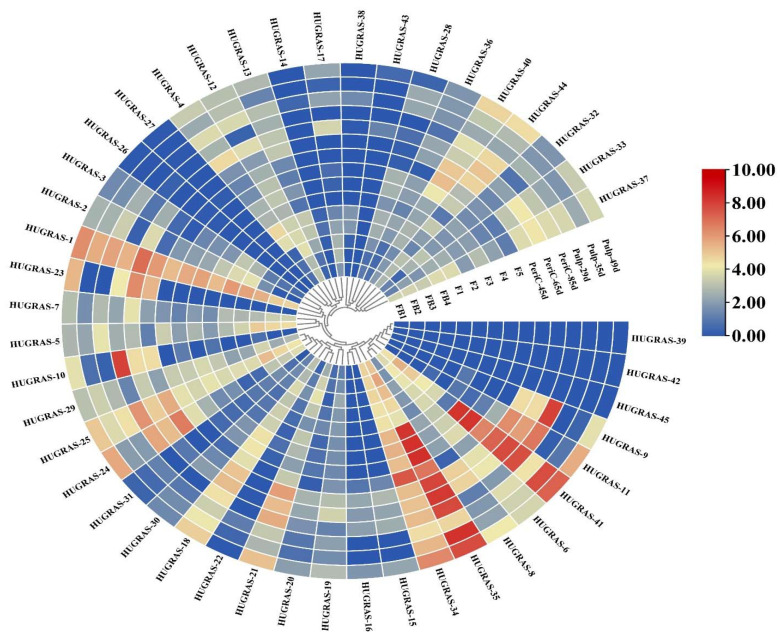
The expression heatmap of *HuGRAS* genes in different pitaya tissues. In this heatmap, 15 rows represent the expressions of different tissues, and 45 columns represent the genes. Four flower bud stages are shown as FB1 to FB4, and five flower stages are shown as F1 to F5. Three pericarp stages are shown as periC-45d, periC-65d and periC-85d, and three pulp stages are shown as pul-29d, pul-35d and pul-49d. Color changes from light blue to dark blue show less or no expression of *HuGRAS* genes. Light yellow to a dark red color shows less expression to a high level of expression of these genes.

**Figure 6 biology-12-00011-f006:**
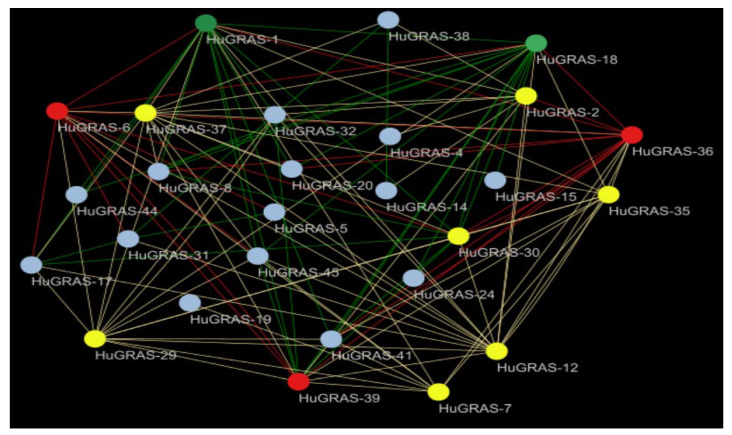
Protein interaction network. Green and red genes are designated as hub genes because they interact with more than 10 genes. Different colors show the interactions of the genes as follows: green (16–20), red (11–15), yellow (6–10) and gray (2–5).

**Figure 7 biology-12-00011-f007:**
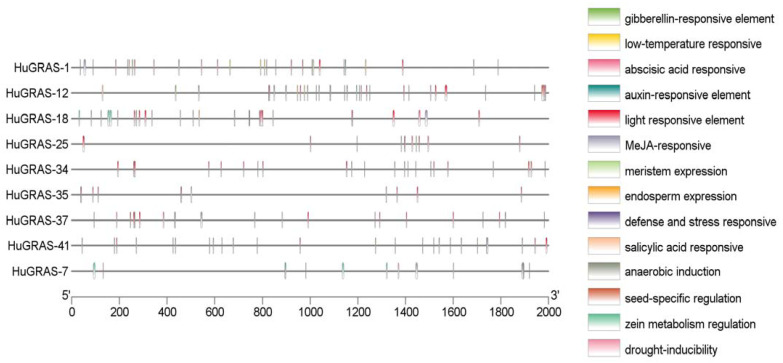
The cis-acting elements of the promoter regions (2000 bp upstream of start codon) of nine *HuGRAS* genes.

**Figure 8 biology-12-00011-f008:**
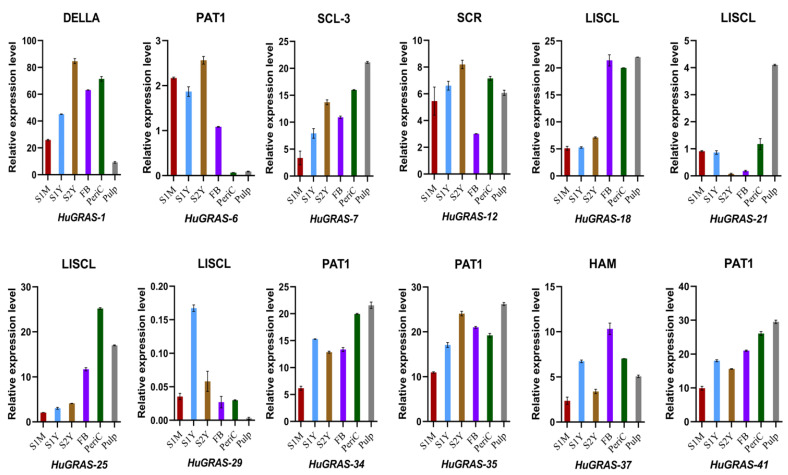
qRT-PCR expression analysis of 12 genes in six tissues of the pitaya (*S. undatus* L.) plant. The X-axis represents the plant tissues, including one-month-old stem (S1M), one-year-old stem (S1Y), two-year-old stem (S2Y), flower bud (FB), pericarp (PeriC) and pulp. Error bars represent the standard deviations for the three replicates.

**Table 1 biology-12-00011-t001:** Physical and chemical properties of *GRAS* genes in pitaya (*S. undatus* L.).

Transcript ID	Renamed ID	Chr. No.	Start–End Position	CDS (bp)	Protein Length (AA)	Protein Mol. Weight (kDa)	pI	GRAVY
*HU10G00709.1*	*HuGRAS-1*	10	8,621,296-8,623,893	1755	584	40.14	5.44	−0.055
*HU11G00778.1*	*HuGRAS-2*	11	78,577,603-78,578,866	1176	391	39.90	6.63	0.105
*HU03G02797.1*	*HuGRAS-3*	3	132,126,059-132,127,531	1473	490	40.39	6.23	−0.027
*HU04G00148.1*	*HuGRAS-4*	4	1,819,377-1,821,914	2055	684	42.26	5.89	−0.221
*HU08G00367.1*	*HuGRAS-5*	8	14,545,649-14,547,799	1350	449	44.77	6.09	−0.13
*HU01G01391.1*	*HuGRAS-6*	1	95,463,256-95,466,959	1623	540	41.20	6.9	−0.101
*HU08G01232.1*	*HuGRAS-7*	8	93,024,447-93,026,894	1392	463	46.25	6.31	−0.086
*HU08G00284.1*	*HuGRAS-8*	8	9,185,291-9,193,945	1533	510	41.50	5.91	−0.067
*HU08G00014.1*	*HuGRAS-9*	8	524,930-530,436	1779	592	41.79	8.7	−0.234
*HU05G01267.1*	*HuGRAS-10*	5	102,083,025-102,085,852	1332	443	46.94	5.99	−0.12
*HU01G01850.1*	*HuGRAS-11*	1	126,187,756-126,191,336	1743	580	41.77	6.3	−0.167
*HU01G00472.1*	*HuGRAS-12*	1	6,212,129-6,216,403	2430	809	38.49	5.76	−0.016
*HU02G03005.1*	*HuGRAS-13*	2	138,909,434-138,911,167	1227	408	39.46	6.44	−0.017
*HU03G01737.1*	*HuGRAS-14*	3	25,880,385-25,881,812	1428	475	46.22	6.57	−0.19
*HU11G01125.1*	*HuGRAS-15*	11	87,396,169-87,400,880	1704	567	41.58	7.1	−0.142
*HU11G01134.1*	*HuGRAS-16*	11	87,665,022-87,669,731	1704	567	41.58	7.1	−0.142
*HU01G00833.1*	*HuGRAS-17*	1	16,765,953-16,767,335	1383	460	34.07	6.36	−0.213
*HU02G01569.1*	*HuGRAS-18*	2	22,485,375-22,488,601	2283	760	42.05	8.6	−0.24
*HU06G00568.1*	*HuGRAS-19*	6	6,285,841-6,289,986	2313	770	42.09	9.09	−0.239
*HU05G01597.1*	*HuGRAS-20*	5	117,157,311-117,159,569	2259	752	42.17	9.46	−0.236
*HU07G02248.1*	*HuGRAS-21*	7	112,127,788-112,125,521	2268	755	42.85	9.24	−0.295
*HU07G02249.1*	*HuGRAS-22*	7	112,130,817-112,133,676	2265	754	42.53	8.96	−0.298
*HU06G00029.1*	*HuGRAS-23*	6	372,597-376,546	1698	565	35.31	6.3	−0.092
*HU02G01571.1*	*HuGRAS-24*	2	22,556,900-22,559,249	2001	666	42.22	8.95	−0.18
*HU07G02246.1*	*HuGRAS-25*	7	112,113,563-112,116,265	1404	467	42.53	9.15	−0.333
*HU07G00272.1*	*HuGRAS-26*	7	2,834,842-2,836,221	1380	459	42.87	5.5	−0.18
*HU11G01529.1*	*HuGRAS-27*	11	92,622,998-92,624,392	1395	464	43.24	5.73	−0.171
*HU06G00358.1*	*HuGRAS-28*	6	3,831,873-3,833,640	1458	485	42.91	5.98	−0.24
*HU02G01572.1*	*HuGRAS-29*	2	22,590,054-22,592,716	2061	686	42.18	9.43	−0.274
*HU02G01570.1*	*HuGRAS-30*	2	22,492,850-22,495,255	2406	801	42.68	8.1	−0.436
*HU02G01573.1*	*HuGRAS-31*	2	22,646,432-22,648,477	2046	681	42.35	9.17	−0.367
*HU05G00466.1*	*HuGRAS-32*	5	8,309,337-8,311,578	1770	589	39.79	6.86	0.143
*HU04G00047.1*	*HuGRAS-33*	4	727,912-731,196	2310	769	39.78	6.33	0.157
*HU06G02537.1*	*HuGRAS-34*	6	123,795,575-123,798,959	2289	762	40.48	7.86	−0.138
*HU08G02295.1*	*HuGRAS-35*	8	106,955,054-106,955,638	585	194	21.76	8.89	−0.617
*HU05G01983.1*	*HuGRAS-36*	5	124,288,256-124,289,817	1455	484	44.22	5.84	−0.235
*HU04G00048.1*	*HuGRAS-37*	4	739,383-742,094	2016	671	41.56	6.59	0.059
*HU06G00376.1*	*HuGRAS-38*	6	4,097,194-4,098,474	1281	426	43.01	5.1	0.006
*HU08G00019.1*	*HuGRAS-39*	8	594,594-594,875	726	241	16.98	6.93	−0.151
*HU06G00283.1*	*HuGRAS-40*	6	594,594-602,896	1371	456	40.78	8.54	0.127
*HU08G02296.1*	*HuGRAS-41*	8	106,955,745-106,956,677	933	310	15.39	7.01	0.115
*HU08G00230.1*	*HuGRAS-42*	8	6,804,389-6,812,672	723	240	16.67	7.95	−0.181
*HU02G03154.1*	*HuGRAS-43*	2	141,447,089-141,448,403	1185	394	37.68	5.87	0.42
*HU05G02245.1*	*HuGRAS-44*	5	126,702,616-126,704,494	1617	538	40.33	8.52	0.201
*HU08G00229.1*	*HuGRAS-45*	8	6,804,079-6,804,372	294	97	95.15	9.56	−0.032

## Data Availability

The data generated and analyzed in this study are available in the Appendix A.

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
