# Peer review of "Genome-Wide Identification and Expression Pattern of the GRAS Gene Family in Pitaya (Selenicereus undatus L.)"

_biology, 2022, doi:10.3390/biology12010011_

Round 1

Reviewer 1 Report

In the current study, authors have identified GRAS family genes in Hylocereus undatus L. and presented their phylogenetic, domains and cis-regulatory elements analysis. However, I found the following problems, which should be addressed before further processing.

General comments:

1.     A few lines e.g. in the abstract are giving a faulty impression about the prediction of function based on qRT-PCR. The high relative expression does not necessarily employ its function in that tissue. But it can be said only that these genes have high expression in these tissues. I found authors went into great pain to interlink functional prediction based on relative expression although the main contribution was genome-wide identification.

2.     Results have been started with 45 genes selection, but why limited to these should be added in results.

3.     Somewhere authors are calling their genes “as genes” and somewhere “as transcription factors”. We need to homogenize. We cannot call any gene as TF until we tested through transcriptional activity assay not only based on domain analysis.

4.     In the discussion, authors should discuss how their results can be helpful for future studies. What is the novelty of the paper, what is new in this study that was not yet explored/examined/explained by others? What are the implications of the results?

Minor comments:

Line 23: what kind of expression analysis; is suggested to remove the sentence ambiguity?

Introduction:

Line 77: italicized the rice (Oryza sativa L.)

Q: Authors should be consistent with scientific names throughout the manuscript. Recommended to choose once and later with a scientific name or generic name or write special circumstances

Line 82: choose a more appropriate word for “discovered” like identified

Q: Suggested changing all (Hylocereus undatus L.) to the short form (H. undatus L.) for other plant species as well throughout the manuscript except being used first time in text

Q: Authors need to verify the working links of the website given in the text

Line 116; delete the “),”

Line 123: add “(“   before http://www.pitayagenomic.com/).

Line 148: clarify the statement “plant buds”

Line 160: clarify the statement RT-PCR or qRT-PCR

Q: provide the info about qRT-PCR gene expression analysis methods in one or two sentences

Results:

Line 171: what is pI?

Line 189-191: remove the sentence ambiguity

Line 270-271: delete gene IDs

Line 312: add “might” played

Line 320: Define “cladode”

Line 321: conform pulp color only one white color?

Line 336: what is the importance of genes which are located on the end region of a chromosome

Line 348: delete “sequences”

Line 366: revise the statement

Line 400: remove the stress tolerance or add in future directions

Reference: Follow the journal recommendations and remove unnecessary punctuation marks “-“ the journal name abbreviations, etc.

Author Response

In the current study, authors have identified GRAS family genes in Hylocereus undatus L. and presented their phylogenetic, domains and cis-regulatory elements analysis. However, I found the following problems, which should be addressed before further processing.

Answer: Thank you very much for your comments! We deeply appreciate your kind words and valuable suggestions, which have improved our manuscript.

General comments:

Query-1. 

A few lines e.g. in the abstract are giving a faulty impression about the prediction of function based on qRT-PCR. The high relative expression does not necessarily employ its function in that tissue. But it can be said only that these genes have high expression in these tissues. I found authors went into great pain to interlink functional prediction based on relative expression although the main contribution was genome-wide identification.

Answer: Thank you very much. We have improved the abstract part as shown below

(Manuscript: L22-L28)

“Among 45 genes, 12 genes were selected due to higher expression in different plant tissues of pitaya plant. In order to verify the RNA-Seq data, 12 genes were selected for qRT-PCR validation. Among the 12 genes, nine GRAS genes exhibited higher expression in different tissues, which indicated the possible role in the growth and development of the pitaya plant. Apart from 45 HuGRAS genes, nine genes are categorized in six subfamilies including DELLA (HuGRAS-1), SCL-3 (HuGRAS-7), PAT1 (HuGRAS-34, HuGRAS-35, HuGRAS-41), HAM (HuGRAS-37), SCR (HuGRAS-12) and LISCL (HuGRAS-18, HuGRAS-25) might be regulating growth and development in pitaya plant.”

Query-2. Results have been started with 45 genes selection, but why limited to these should be added in results.

Answer: Thank you so much. We have modified the MS text as shown in line-180.

We can only retrieve the domains on 45 Hylocereus undatus genes in the pitaya genome. That’s why we selected these genes as shown in figure 1.

Query-3.     

Somewhere authors are calling their genes “as genes” and somewhere “as transcription factors”. We need to homogenize. We cannot call any gene as TF until we tested through transcriptional activity assay not only based on domain analysis.

Answer: Thank you so much for indicating the mistake. We have improved in the MS.  

Query-4.    

 In the discussion, authors should discuss how their results can be helpful for future studies. What is the novelty of the paper, what is new in this study that was not yet explored/examined/explained by others? What are the implications of the results?

Answer: Thank you so much. We have included the text in the manuscript discussion part Line-398 to Line-408.

Query-5 Minor comments:

Line 23: what kind of expression analysis; is suggested to remove the sentence ambiguity?

Answer: Thank you very much for your suggestions! We have checked and modified the text and nomenclature. Currently, all the figures and nomenclature are listed in order in our manuscript (Section 3.3 highlighted in red color,). 

Introduction:

Line 77: italicized the rice (Oryza sativa L.)

Answer: The word is italicized as in Line 76.

Q: Authors should be consistent with scientific names throughout the manuscript. Recommended to choose once and later with a scientific name or generic name or write special circumstances

Answer: Thank you very much. We have checked it throughout the manuscript.

Line 82: choose a more appropriate word for “discovered” like identified

Answer: The sentence is modified (Line 81-82.).

Q: Suggested changing all (Hylocereus undatus L.) to the short form (H. undatus L.) for other plant species as well throughout the manuscript except being used first time in text

Answer: We have modified throughout the manuscript.

Q: Authors need to verify the working links of the website given in the text

Answer: Thank you very much. We have checked it throughout the manuscript.

Line 116; delete the “),”

Answer: The sign “),” is deleted.

Line 123: add “(“   before http://www.pitayagenomic.com/).

Answer: The website link is added in the MS.

Line 148: clarify the statement “plant buds”

Answer: Thank you very much. We have modified it. It is a flower bud, not a plant bud (L145).

Line 160: clarify the statement RT-PCR or qRT-PCR

Answer: It is qRT-PCR. The sentence is modified in Line 158.

Q: provide the info about qRT-PCR gene expression analysis methods in one or two sentences

Answer: Thank you very much for pointing out the mistake. We have added the citation and sentence as well.  

Line 171: what is pI?

Answer:It is an isoelectric point and the sentence is modified as mentioned in Line 167.

Line 189-191: remove the sentence ambiguity

Answer: The sentence is changed as in Line 187-191.

Line 270-271: delete gene IDs

Answer: The gene IDs have been deleted.

Line 312: add “might” played

Answer: It is added to the text as in Line 310.

Line 320: Define “cladode”

Answer: It is defined in Line 318-320 as “Pitaya (H. undatus L.) as a tropical fruit typically a cactus, evergreen, annual plant cladode (a modified stem replacing the leaves for photosynthesis function), resembles and function as a leaf”.

Line 321: conform pulp color only one white color?

Answer: The species name of white pulp pitaya is added in the sentence.

Line 336: what is the importance of genes which are located on the end region of a chromosome

Answer: The physical location of the gene and its interacting expression is still largely unclear. However, it is perceived as the different parts of the nuclear periphery have a distinct role in transcriptional regulation. But Fraser and Bickmore, 2007 presented an emerging idea that the clustering of the gene in transcription hot spots contributes to their efficient regulation and expression. It is also linked to the replication process. However, gene position linked with gene function is affected by multiple components and It is not easy to conclude a universal rule.

Line 348: delete “sequences”

Answer: The sentence is modified.

Line 366: revise the statement

Answer: The sentence is revised.

Line 400: remove the stress tolerance or add in future directions

Answer: The sentence is modified.

Reference: Follow the journal recommendations and remove unnecessary punctuation marks “-“ the journal name abbreviations, etc.

Answer: Thank you very much. All references are modified as per journal format.

Reviewer 2 Report

In the manuscript “Genome-Wide Identification and Expression Pattern of GRAS Gene Family in Pitaya Fruit (Hylocereus undatus L.)Zaman et al described the expression of GRAS family and identified the candidate genes regulating the growth and development of the pitaya fruit. The authors wrote a well-prepared article. But still, I have suggestions to improve it. 

L2: Scientific name Hylocereus undatus L. write after “Pitaya”, not after fruit. 

Line-10, ambiguous statement, modify it. 

L13: Among the 45 HuGRAS family members” change to “Among these HuGRAS family members

Line-37: Add reference here. 

L40: “Pitaya gained the attention” change to “Pitaya fruit gained the attention”

L51: Pitaya to pitaya. 

L72: “Genome wide” change to Genome-wide. See thoroughly in MS. 

L94: Though dragon fruit is also a synonymous word for pitaya. But use one, throughout the manuscript. Change dragon fruit to pitaya fruit. 

L113: change “Soybean” to soybean

L122: Delete the word FASTA file. No need to write the format of the file. 

L136: Delete repetition of the word “three stages”. Modify it. 

L167: The heading style should be same throughout the MS. 

L203:  Revise the figure legend. 

L249: No need to cite the website here again. It is already mentioned in materials and methods section. 

L249-250: Rewrite this sentence. 

L305: correct grammar mistake

L311-313: Revise the sentence. 

L315: Delete the word “Selected”

L334: “Hylocereus undatus genome” change to “pitaya”.

L343: “335 sequences of six other species including maize, soybean, Medicago
truncatula, rice, Arabidopsis and tomato” change to “335 sequences of GRAS proteins of maize, soybean, Medicagotruncatula, rice, Arabidopsis and tomato”

L347-L348: Protein sequences and expression
profile of the pitaya different tissues” change to “Different expression profile of pitaya tissues”

L349: Delete the word “analysis” repeated two time 

L354-359: Delete the gene IDs. Because the author already abbreviated them in the table. Line-369: It’s just a prediction that DELLA interacts with PIF family members. You have not proven this with any analysis or experiment in this MS. So, I think, the author should consider it to remove the sentence.

Author Response

In the manuscript “Genome-Wide Identification and Expression Pattern of GRAS Gene Family in Pitaya Fruit (Hylocereus undatus L.)”, Zaman et al described the expression of GRAS family and identified the candidate genes regulating the growth and development of the pitaya fruit. The authors wrote a well-prepared article. But still, I have suggestions to improve it. 

Thank you very much for pointing out the mistakes in the manuscript. We deeply appreciate for your kind words and valuable suggestions, which have improved our manuscript.

L2: Scientific name Hylocereus undatus L. write after “Pitaya”, not after fruit. 

Answer: Thank you very much. We have revised the title of the MS.

Line-10, ambiguous statement, modify it.

Answer: The statement has been revised.

L13: Among the 45 HuGRAS family members” change to “Among these HuGRAS family members

Answer: It is changed.

Line-37: Add reference here. 

Answer: Reference is added as mentioned in Line 38.

L40: “Pitaya gained the attention” change to “Pitaya fruit gained the attention”

Answer: The statement has been revised.

L51: Pitaya to pitaya.

Answer: It is modified.  

L72: “Genome wide” change to Genome-wide. See thoroughly in MS. 

Answer: This is changed through out the manuscript.

L94: Though dragon fruit is also a synonymous word for pitaya. But use one, throughout the manuscript.

Answer: All MS text, headings and subheadings have been revised and replaced with the word pitaya.

Change dragon fruit to pitaya fruit. 

Answer: It is changed.

L113: change “Soybean” to soybean

Answer:It is changed.

L122: Delete the word FASTA file. No need to write the format of the file. 

Answer: It is deleted.

L136: Delete repetition of the word “three stages”. Modify it. 

Answer: It is deleted.  

L167: The heading style should be same throughout the MS.

Answer: It is modified.   

L203:  Revise the figure legend.

Answer: Figure legend has been revised as mentioned below

Figure 2. Characterized sequences of six species (maize, soybean, Arabidopsis, Medicago truncatula, tomato and rice) were used to draw the phylogenetic tree with the pitaya GRAS genes. The GRAS proteins were divided into 9 subfamilies exhibited with different colors including PAT1 (red), SHR (navy blue), LISCL (light green), HAM (light orange/Wheat color), SCR (Violet), RGL (pea green), LAS (teal), DELLA (magenta pink) and SCL3 (gray).

L249: No need to cite the website here again. It is already mentioned in the materials and methods section. 

Answer: The weblink is deleted

L249-250: Rewrite this sentence. 

Answer: The sentence is modified as mentioned in Line 248-251.  

L305: correct grammar mistake

Answer: Thank you very much. The sentence is modified  as mentioned in Line 301-303.

L311-313: Revise the sentence. 

Answer: It is revised.  

L315: Delete the word “Selected”

Answer: It is deleted.  

L334: “Hylocereus undatus genome” change to “pitaya”.

Answer: It is changed.

L343: “335 sequences of six other species including maize, soybean, Medicago
truncatula, rice, Arabidopsis and tomato” change to “335 sequences of GRAS proteins of maize, soybean, Medicagotruncatula, rice, Arabidopsis and tomato”

Answer:The Sentence is modified as mentioned in Line 345-347.  

L347-L348: Protein sequences and expression
profile of the pitaya different tissues” change to “Different expression profile of pitaya tissues”

Answer: It is changed.

L349: Delete the word “analysis” repeated two time 

Answer: It is deleted.

L354-359: Delete the gene IDs. Because the author already abbreviated them in the table. Line-369: It’s just a prediction that DELLA interacts with PIF family members. You have not proven this with any analysis or experiment in this MS. So, I think, the author should consider it to remove the sentence.

Answer: Thank you very much for the suggestions. All gene IDs have been deleted.

Reviewer 3 Report

The MS "Genome-Wide Identification and Expression Pattern of GRAS Gene Family in Pitaya Fruit (Hylocereus undatus L.)" by Zaman et al. covers interesting research. In summary, the result of this study provides molecular insights into the growth and development of pitaya plant including GRAS family gene function in protein interactions, signaling pathway regulations, and expression patterns. The manuscript is well organized and can be accepted after clarification/consideration of the following points:

1. Line 27-28: Authors have given example of some plants (maize, rice, soybean, tomato, Medicago and Arabidopsis), morphologically they are differentiated into proper leaf stem and other plant part however, the morphology of Pitaya plant is more complex (not differentiated into proper leaf, stem). Therefore, please clarify among these six gene families DELLA (HU10G00709.1), SCL-3 (HU08G01232.1), PAT1 (HU06G02537.1, HU08G02295.1, HU08G02296.1), HAM (HU04G00048.1), SCR (HU01G00472.1) and LISCL (HU02G01569.1, HU07G02246.1) which gene families are responsible for Pitaya plant morphology in more detail. 

2. Line 325-330: Please provide in depth the GRAS gene and phytohormone (Gibberellic acid, Auxins) relationship & its role in respect to the growth and development including pathway. 

3. Line 365-368: Please clarify the DELLA protein & gibberellic acid level in experiment which are not justified by previous workers.

4. In references all the Botanical names are given in normal font which is incorrect. Botanical names should be in ITALIC FONT. Please carefully check each reference and CORRECT all the botanical names in ITALIC Font.

5. In references Journal names are not uniform. Most of the Journal names are in full, however some are in abbreviated form (please see: 8, 13, 17,19, 23,24, 26, 31, 33, 34, 36,37, 47, 50, 54, 55,56, 57, 59, 61, 65, 66, 67). Please correct all the abbreviated Journal names in FULL to maintain uniformity.

Author Response

The MS "Genome-Wide Identification and Expression Pattern of GRAS Gene Family in Pitaya Fruit (Hylocereus undatus L.)" by Zaman et al. covers interesting research. In summary, the result of this study provides molecular insights into the growth and development of pitaya plant including GRAS family gene function in protein interactions, signaling pathway regulations, and expression patterns. The manuscript is well organized and can be accepted after clarification/consideration of the following points:

Answer: Thank you very much for evaluating the manuscript.  We deeply appreciate your valuable suggestions, which have improved our manuscript. 

Query 1: Line 27-28: Authors have given example of some plants (maize, rice, soybean, tomato, Medicago and Arabidopsis), morphologically they are differentiated into proper leaf stem and other plant part however, the morphology of Pitaya plant is more complex (not differentiated into proper leaf, stem). Therefore, please clarify among these six gene families DELLA (HU10G00709.1), SCL-3 (HU08G01232.1), PAT1 (HU06G02537.1, HU08G02295.1, HU08G02296.1), HAM (HU04G00048.1), SCR (HU01G00472.1) and LISCL (HU02G01569.1, HU07G02246.1) which gene families are responsible for Pitaya plant morphology in more detail. 

Answer:  The GRAS gene family is divided based on its structure into the following subfamilies including DELLA, HAM, LAS, LISCL, RGL, PAT1, SCR, SCL3 and SHR. Though these subfamilies are mostly reported in those plant species which have well-differentiated parts of the plant. But the pitaya plant develops a cladode which is also a modified form of the stem, which functions as a leave for photosynthesis. So we can consider cladode as a leave of the plant.

In result section-3.8, we have elaborated the relative expression of each gene belonging to these six sub-families as below. Thanks for understanding.

“HuGRAS-1 gene categorized in DELLA subfamily significantly expressed across the tissues including stem, FB and Pericarp. But relatively weaker expression was observed in pulp of the fruit. Among the PAT1 subfamily members, HuGRAS-34, HuGRAS-35, HuGARS-41 exhibited strong expression in plant tissues as compared to HuGRAS-6 which exhibited weakly expression in pericarp and pulp. HuGRAS-7, which belongs to the SCL-3 subfamily, was expressed at a low level in one-month-old stem cells but is abundant in other tissues. HuGRAS-12, a gene categorized in the SCR subfamily was expressed at a higher level in other tissues than the flower buds. HuGRAS-21 and HuGRAS-29 members of the LISCL subfamily were expressed at lower level than the HuGRAS-18 and HuGRAS-25 grouped in a same subfamily, which were expressed at a higher level in the flower buds, pericarp and pulp of the pitaya plant. HuGRAS-37 is grouped into HAM subfamily, highly expressed in flower buds but weakly expressed in one month old stems. Nine genes exhibited higher expression level which were categorized into six subfamilies and might play a key role in the growth and development of the pitaya plant”.  

Query 2:. Line 325-330: Please provide in depth the GRAS gene and phytohormone (Gibberellic acid, Auxins) relationship & its role in respect to the growth and development including pathway. 

Answer:  Thank you very much. The function of GRAS genes and subfamilies is explained in terms of growth and development in the introduction part of the manuscript from L50-71. However, specifically for the phytohormone, it is explained in the discussion part from Line 360. Thanks for understanding.

Query 3:. Line 365-368: Please clarify the DELLA protein & gibberellic acid level in an experiment which are not justified by previous workers.

Answer:  Thank you very much. DELLA proteins are the key negative regulator of the Gibberellic acid signaling pathway, and it mediates the synergistic regulation of gibberellin and light signal by interacting with PIF protein. DELLA proteins participate in signal transmission and regulation of various hormones as a core role in the plant hormone regulation network. In our cis-acting elements, it was found that gibberellic acid was found abundantly in GRAS genes. So, these elements are playing a key role in the growth and development of the pitaya plant. Thanks for understanding. As it is explained in L362-371.

Query 4:. In references all the Botanical names are given in normal font which is incorrect. Botanical names should be in ITALIC FONT. Please carefully check each reference and CORRECT all the botanical names in ITALIC Font.

Answer:  Thank you very much. We have modified all of these in the MS text.

Query 5:. In references Journal names are not uniform. Most of the Journal names are in full, however some are in abbreviated form (please see: 8, 13, 17,19, 23,24, 26, 31, 33, 34, 36,37, 47, 50, 54, 55,56, 57, 59, 61, 65, 66, 67). Please correct all the abbreviated Journal names in FULL to maintain uniformity.

Answer:  Thank you very much. All references are modified as per journal format.

Reviewer 4 Report

The article titled as “Genome-Wide Identification and Expression Pattern of GRAS Gene Family in Pitaya Fruit (Hylocereus undatus L.) by Zaman et al summarized the role of the GRAS gene family and identified some potential candidate genes regulating the growth and development in pitaya. The author reported the expression pattern of GRAS genes in the pitaya plant. Overall, this is a timely and well-written article on the pitaya plant. It can be accepted for publication in Biology-MDPI journal.

In addition to that, I have suggested some changes which can improve the article for publication in the journal. Some examples are given below for consideration.

1, The word “Pitaya” should be written as pitaya.

2. In some cases, the author write genome wide, at some point, it is written as genome-wide.

3, Author changed the gene IDs and renamed them as HuGRAS-1 to HuGRAS-45. But in some sections, the author wrote both name including gene ID and given name.

4, In the 3rd paragraph of the discussion, author discussed about the genes function in respect of GRAS gene subfamilies. In this section can add more points because these subfamilies are elaborated in other species. More references can be added here.

5, Line-44-Line-45: Add citations for each transcription factor. Line-59 to Line-60: Mention all functions of the GRAS gene family here as the author write in the abstract 1st line also. Line-86: author predicted 12 genes as mentioned in the abstract but in Line 86, the author mentioned 10 genes. Please correct the statement. Line-99: In line 83, the author has used the word maize, here used corn. Use one throughout the manuscript. Line-102: Change “Proteins” to “proteins” Line-108: Modify the sentence. Line-226: Delete the gene IDs. Line-229 to Line-230: Modify the sentence structure. Line-236 to Line-238. Improve the figure legend. Line-367: Delete Gene ID. Please set the references as per Biology MDPI format.

Author Response

The article titled as “Genome-Wide Identification and Expression Pattern of GRAS Gene Family in Pitaya Fruit (Hylocereus undatus L.) by Zaman et al summarized the role of the GRAS gene family and identified some potential candidate genes regulating the growth and development in pitaya. The author reported the expression pattern of GRAS genes in the pitaya plant. Overall, this is a timely and well-written article on the pitaya plant. It can be accepted for publication in Biology-MDPI journal.

Answer: Thank you very much. We deeply appreciate your kind words and valuable suggestions, which have improved our manuscript.

In addition to that, I have suggested some changes which can improve the article for publication in the journal. Some examples are given below for consideration.

Query 1, The word “Pitaya” should be written as pitaya.

Answer: All common names have been modified.

Query  2. In some cases, the author write genome wide, at some point, it is written as genome-wide.

Answer: Thank you very much. It is changed throughout the manuscript.

Query  3, Author changed the gene IDs and renamed them as HuGRAS-1 to HuGRAS-45. But in some sections, the author wrote both name including gene ID and given name.

Answer: Thank you very much. We have deleted the gene IDs from text of the MS.

Query  4, In the 3rd paragraph of the discussion, author discussed about the genes function in respect of GRAS gene subfamilies. In this section can add more points because these subfamilies are elaborated in other species. More references can be added here.

Answer: Thank you very much. Function GRAS gene subfamilies is mentioned in the MS text. As the function of the DELLA subfamily is elaborated in Line 360. Function of PAT1 is elaborated in Line 372. The function of SCL3 is explained in Line-375, HAM in Line 382, SCR in Line 383. Likewise, their overview is also added in the introduction part of the manuscript. Thanks for understanding.  

Queries

5, Line-44-Line-45: Add citations for each transcription factor. Line-59 to Line-60: Mention all functions of the GRAS gene family here as the author write in the abstract 1st line also (Thank you). Line-86: author predicted 12 genes as mentioned in the abstract but in Line 86, the author mentioned 10 genes (Thank you). Please correct the statement. Line-99 (Thank you).: In line 83, the author has used the word maize, here used corn (Thank you).. Use one throughout the manuscript. Line-102: Change “Proteins” to “proteins” (Thank you).  Line-108: Modify the sentence. (Thank you).  Line-226: Delete the gene IDs. Line-229 to Line-230: Modify the sentence structure. Line-236 to Line-238. Improve the figure legend. Line-367: Delete Gene ID (Thank you).. Please set the references as per Biology MDPI format.

Answer: Thank you very much for your valuable time in reviewing the article. We have modified our manuscript as per your valuable suggestions.  

Round 2

Reviewer 1 Report

Authors have revised the manuscript substantially, however following comments should be addressed before acceptance.

Figure 3 legends have separate explaination for A and B. But i cannot see any panels with the name A and B in Figure 3.

Figure 7 legneds represent the cis-acting elements of all 45 genes, but i cannot see only few genes in figures 7.

Line 92, Change were to was

Line 104, change conserve to conserved

Line 148, has grammar mistake

Line 199, change being to is

Author Response

Authors have revised the manuscript substantially; however following comments should be addressed before acceptance.

Thank you very much for your valuable time in reviewing the article. We have modified our manuscript as per your valuable suggestions.  

Query-1:  Figure 3 legends have a separate explanation for A and B. But i cannot see any panels with the name A and B in Figure 3.

Thank you very much for pointing out the mistake. We have labeled  figure 3.

Query-2:  Figure 7 legends represent the cis-acting elements of all 45 genes, but i cannot see only few genes in Figure 7.

Thank you very much. We have modified the figure legends. Thanks again.

Query-3:   Line 92, Change were to was

Text is modified.

Query-4:   Line 104, change conserve to conserved

Text is modified.

Query-5:   Line 148, has grammar mistake

The sentence is revised.

Query-6:  Line 199, change being to is

We improved the sentence text.

Reviewer 4 Report

Authors have significantly improved the manuscript. 

Author Response

Authors have significantly improved the manuscript. 

We deeply appreciate for your kind words and valuable suggestions, which have improved our manuscript.